# First Report of Hemp Fusarium Wilt Caused by *Fusarium oxysporum* in Croatia

**DOI:** 10.3390/plants12183305

**Published:** 2023-09-18

**Authors:** Tomislav Duvnjak, Karolina Vrandecic, Aleksandra Sudaric, Jasenka Cosic, Tamara Siber, Maja Matosa Kocar

**Affiliations:** 1Department of Industrial Plants Breeding and Genetics, Agricultural Institute Osijek, 31000 Osijek, Croatia; 2Department of Phytomedicine, Faculty of Agrobiotechnical Sciences Osijek, Josip Juraj Strossmayer University of Osijek, 31000 Osijek, Croatia; 3Center of Excellence for Biodiversity and Molecular Plant Breeding, Faculty of Agriculture, University of Zagreb, Svetosimunska Cesta 25, 10000 Zagreb, Croatia

**Keywords:** hemp, *Fusarium oxysporum*, wilt, Croatia

## Abstract

Wilted hemp (*Cannabis sativa* L.) plants were observed in August 2019 in commercial fields around Osijek, Croatia. Plants and roots with disease symptoms were collected. The single-spored isolates produced septate cottony white to light pink aerial mycelium and purple undersurface on potato dextrose agar (PDA). Smooth and hyaline hyphae were branched and septate. Macroconidia were fusiform to sickle-shaped with foot-shaped basal cells, elongated apical cells and three to five septa. Sequencing of the internal transcribed spacer and the partial elongation factor 1-α gene identified the species as *Fusarium oxysporum*. Artificial infection fulfills Koch’s postulates, producing plants which show stunted growth and wilt symptoms similar to those observed in the commercial fields. Control seedlings remained symptomless and healthy. To the best of our knowledge, this is the first report of hemp Fusarium wilt causing *F. oxysporum* in Croatia. Considering that *F. oxysporum* has been reported in main field crops in Croatia, the presence of this pathogen could cause economically significant hemp production decreases, especially in humid and cold springs and susceptible varieties.

## 1. Introduction

Hemp (*Cannabis sativa* L., Cannabaceae) is considered one of the oldest crops known to man [1]. It is estimated that its use dates back to 10,000 years ago [2] and a hypothesis of co-evolution of the genus *Cannabis* with the human species has been postulated [3]. The hemp plant is native to India and Persia, although it has been cultivated in nearly all temperate and tropical countries [4,5]. Hemp is a versatile herbaceous crop that has been used for fiber, food and medicinal purposes [6,7,8,9]. Hemp is an inexpensive and available bast natural fiber [10]. Fiber production from hemp has been conducted over many centuries, for end uses ranging from textiles and papers (hemp paper was used in the first copies of the Bible) to ropes and sails [11]. Cannabis has been widely cultivated due to its industrial [12], ornamental [13], nutritional [14], medicinal and recreational [15] potential. From regulatory and application perspectives, cannabis plants are categorized based on the level of Δ9-tetrahydrocannabinol (THC), one of the most important phytocannabinoids [16]. Plants are generally classified and regulated as industrial hemp if they contain less than 0.3% THC in the dried flower (this level varies by country) or drug-type with more than this threshold [17].

The cultivation of hemp dates back to China around 2700 BC and is believed to have then expanded across Asia, making its way to Europe 2000–2200 years ago [3,4]. As a multi-use crop, hemp is considered one of the oldest plants cultivated to provide nutritional and medicinal benefits [7,18]. Additionally, hemp provides raw materials for a large number of traditional and innovative industrial applications [1]. In recent years, the interest in investigating the potential use of industrial hemp in food and nutraceuticals has been growing [19]. Pressure from weeds, insects and diseases are among the major challenges in the production of *C. sativa* [20]. *Fusarium* species can parasitize a wide range of plants, including vegetables, flowers, field crops (cotton, hemp and tobacco), herbaceous perennial ornamentals and plantation crops (banana, plantain, coffee, sugarcane) [21]. Additionally, they can parasite vertebrates, insects, humans, or even other fungi. Conducted mycological analysis of seed and grain of many crops from 2002 to 2008 in Croatia revealed the presence of *Fusarium* species on wheat, barley, oat, triticale, maize, tobacco, bean, pea, soybean, lupin, vetch, alfalfa, clover, flax, beet, spinach and lettuce seed and grain [22]. *F. oxysporum* has been confirmed as a cause of wilting soybeans in Croatia [23].

All parts of the plant at all growth stages are infected by one or more pathogens while pathogens in the genus *Fusarium* are among the most destructive pathogens of *C. sativa* [24]. Several *Fusarium* species attack hemp crops and cause damping off, including *Fusarium solani*, *Fusarium oxysporum* [25] and less frequently *Fusarium sulphureum*, *Fusarium avenaceum*, *Fusarium graminearum*, *Fusarium culmorum*, *Fusarium avenaceum* and *Fusarium fujikuroi* [24]. The time of infection and vegetative growth phases result in decreased plant quality or even total plant loss. Species of *Fusarium* also cause significant postharvest losses and produce mycotoxins that further limit the value of the crop [26]. Since, for many countries, the legal status of the plant changed from crop to illegal drug (1950s) and back again in the 2000s [27], the bulk of literature on *C. sativa* and *Fusarium* is written in two distinct time periods (prior to 1960 and after 1990). There are some exceptions to this bimodal distribution, particularly publications reporting the use of *F. oxysporum* f. sp. *cannabis* as a biological control of the plants [28].

*F. oxysporum* is a common fungal pathogen that causes wilt disease in nurseries and in field cultivation and causes high losses [29]. Some of these fungi also cause cotyledon drop [30]. In mature plants these pathogens cause Fusarium root rot, stem canker and wilt [31]. Fusarium wilt is the name of a disease caused by two microscopic fungi that infect hemp roots and then move systemically through the plant. Symptoms of Fusarium wilt begin as small, dark, irregular spots on lower leaves [32] and pathogen-affected leaves suddenly become chlorotic (yellow). Wilt symptoms begin with an upward curling of leaf tips. Wilted leaves dry to a yellow-tan color and hang on the plant without falling off (Figure 1).

Petioles, branches and sometimes even the stalk may droop while stems also turn yellow-tan. Cutting into wilted stems reveals a reddish-brown discoloration of xylem tissue. Pulled-up roots show no external symptoms. Fusarium wilt is a warm-weather disease, and the optimal temperature for fungal growth is 26 °C. Disease symptoms may not become evident until the advent of hot summer temperatures [33]. *Fusarium* conidia spread via splashed rain and water runoff. *Fusarium* spp. can live saprophytically on roots, stems, leaves, flowers and seeds of diseased and dead plants [34]. The fungus can survive on seed (internal and external) or as spores or mycelium in dead or infected tissues. The pathogen is disseminated by air, equipment and water [21].

Fusarium wilts are of worldwide importance and disease severity is favored by warm climates and warm soil temperatures [24]. Wilt development is favored by air temperatures of 28 °C, low levels of nitrogen and phosphorus, elevated levels of potassium, low soil pH, short day length and low light intensity [21]. In other hosts, the virulence of *Fusarium* can be enhanced with ammonium nitrogen and decreased by nitrate nitrogen [35]. Many species of *Fusarium* co-exist with their hosts as endophytes and protect the host against pathogens and *F. oxysporum* has been reported as endophytic in *C. sativa* [26].

*F. oxysporum* is typically soil-borne and is the most economically important and the most commonly isolated species of *Fusarium* [36]. Due to the large numbers of spores produced, oil infested with *Fusarium* may remain so indefinitely [37]. Seed-borne infections lay dormant until seedlings sprout the following spring. Mycoherbicide researchers reported that spore-coated hemp seeds effectively spread *F. oxysporum* through the soil [38]. 

Fusarium wilt reduces fiber quality and fiber yield and reduces seed production and seed quality. Seeds infested by the wilt fungus should not be used for human consumption, or for cultivation or breeding purposes. Mycotoxins, produced by many species of *Fusarium*, are of greatest agronomic importance and limit the food and feed supply [24]. Mycotoxins can create serious health problems in humans and animals [39].

## 2. Materials and Methods

### 2.1. Fungal Strain Collection, Isolation and Growth Conditions

Numerous wilted hemp plants (~15%) were observed in August 2019 in commercial fields in Vladislavci, around Osijek (45.4646950° N, 18.5674770° E), Croatia. Vladislavci is located about 12 km air distance southwest of Osijek, at an altitude of 85.69 m. The average amount of precipitation for Osijek is 702.6 mm. The rainiest month is June (81.5 mm), and the least precipitation is on average in February (42.2 mm). The average total temperature is 11.8 °C, the coldest month is January with an average air temperature of 0.6 °C, while the warmest month is July with an average air temperature of 22.4 °C (all data are 30-year average, 1991–2020).

Several pieces (the stem and the root) of the infected hemp plants (2- to 3 cm pieces) were sampled with sterile blades, placed in sterile plastic bags and transferred to the laboratory. Small pieces of tissue (0.5–1 cm) were treated with 1% sodium hypochlorite solution for 5 min and rinsed three times with sterile water. Pieces were left to dry under hood flow and placed on half-strength potato dextrose agar (PDA, Biolife Italiana, Milan, Italy) amended with streptomycin stock solution (Sigma-Aldrich, St. Louis, MO, USA) to inhibit bacterial growth. 

The culture plates were incubated at 25 ± 2 °C for 7 days and purified to obtain single-spore isolates by sterile needles using a Nikon E400 Eclipse stereomicroscope, Tokyo, Japan under M18 laminar flow hood. The single-spore isolates produced septate cottony white to light pink aerial mycelium and purple undersurface on PDA (Figure 2a). Smooth and hyaline hyphae were branched and septate. Morphological identification was carried out according to The *Fusarium* Laboratory Manual [40]. 

### 2.2. Fungal Strain Molecular Isolation

#### 2.2.1. Internal Transcribed Spacer Region Analysis

The *Fusarium oxysporum* isolate was identified at the species level by ITS rDNA sequencing. Mycelium was harvested from the colony surface by a sterile medicine spoon [41]. The primers ITS4/ITS5 (ITS4 5′-TCCTCCGCTTATTGATATGC-3′ and ITS5 5′-GAAAGTAAAAGTCGTAACAAGG-3′) [41,42] were used in the following reaction mixture: ~150 mg of fresh mycelium, 2 μL of 20 mg μL^−1^ bovine sieroalbumin solution; 1.5 μL of 50 U μL^−1^ Taq polymerase solution; 5 μL Buffer 10×; 1 μL od 10 mM dNTP, 4 μL of 50 Mn MgCl_2_ solution; 2 μL ITS4 Primer forward; 2 μL ITS5 primer reverse; and sterile distilled water up to 50 μL. Polymerase chain reaction (PCR) amplification was carried out in a thermal cycler (SimpliAmp^TM^ Thermal Cycler—Applied Biosystems, Waltham, MA, USA) with the following program: 95 °C for 3 min, followed by 34 cycles of denaturation at 95 °C for 30 s, annealing at 54–57 °C for 30 s, extension at 72 °C for 1 min and final extension 72 °C for 8 min. Sequencing was carried out by Microsynth AB Company (Balgach, Switzerland), starting from the solution of amplicons obtained by PCR checked on 1.5% agarose gel.

The ITS sequences were compared with those available in the National Center for Biotechnology Information (NCBI, http://www.ncbi.nlm.gov/; accessed on 25 April 2023) genetic database by using the Basic Local Alignment Tool (BLAST) algorithm and using only sequence identity values above 99%. 

#### 2.2.2. Translation Elongation Factor 1-α Gene Analysis

A second PCR analysis was performed using fungi-specific *tef* pair of primers EF1 (5′ ATGGGTAAGGAAGGACAAG 3′) and EF 2 (5′ GGAGAGTACCAGTGCATCAT 3′) [43]. Total DNA from isolate HFox1 was extracted with OmniPrep™ for Fungi PCR Kit (G-Biosciences Co., St. Louis, MO, USA). Amplification of the translation elongation factor 1α was performed with primer pair EF1 TEF -1α/EF2 [43] using EmeraldAmp MAX PCR Master Mix (Takara Bio USA, Inc., San Jose, CA, USA) in a reaction mixture containing 0.2 µM µL of each primer and 2 µL of undiluted genomic DNA. Polymerase chain reaction (PCR) was conducted using MiniAmp Plus Thermal Cycler (Applied Biosystems, Waltham, MA, USA) with the following conditions: an initial denaturation step at 95 °C for 5 min, 35 amplification cycles with denaturation at 94 °C for 30 s, primer annealing at 55 °C for 45 s and elongation at 72 °C for 1 min 30 s, followed by a final elongation step at 72 °C for 10 min. Obtained PCR products were visualized on 1% agarose gel pre-stained with SYBR Safe DNA Gel Stain, and documented using the Kodak EDAS 290 system with a UV transilluminator (UVITEC). PCR products were purified Wizard^®^ SV Gel and PCR Clean-Up System (Promega Corporation, Madison, WI, USA), and sequenced by Macrogen Europe (Amsterdam, The Netherlands) on both strands with the same primers used for amplification. The sequence assembling was performed with DNADynamo Software (BlueTractorSoftware, North Wales, UK), and nucleotide sequence data was deposited in GenBank under accession number OR149071. The result was confirmed with an additional primer pair Fa+7/Ra+6 [43] which gave an identical nucleotide sequence for the translation elongation factor 1α of isolate HFox1 Amplification with primers Fa+7/Ra+6 was the same as described for EF1 TEF -1α/EF2 except that primer annealing was at 67 °C for 1 min. 

The EF sequences were compared with those available in the National Center for Biotechnology Information (NCBI, http://www.ncbi.nlm.gov/; accessed on 12 July 2023) genetic database by using the Basic Local Alignment Tool (BLAST) algorithm and using only sequence identity values above 99%.

### 2.3. Phylogenetic Analysis

This analysis involved 46 nucleotide sequences for ITS and TEF (Table 1). Both, ITS and TEF sequence of *HFox1* isolate was submitted to National Center for Biotechnology Information (NCBI)—*GenBank* accession numbers OM475708 and OR149071. The bioinformatics algorithm basic local alignment search tool (BLASTn) program [44] was used for sequence analysis. 

Using the program Seaview [45], we combined both sequences to obtain a concatenated multilocus dataset. Phylogenetic trees were constructed using ClustalW sequence alignment and evolutionary analyses were conducted in MEGA11 [46]. Bayesian analyses were performed in MrBayes v.3.2.7a [47]. The Markov Chain Monte Carlo sampling (MCMC) analyses were conducted with six simultaneous Markov chains. They were run for 120,000 generations; sampling the trees at every 100th generation. The resulting phylogenetic trees (Figure 3) were drawn using FigTree v.1.4.4 [48]. 

### 2.4. Pathogenicity Test

To confirm their pathogenicity, isolate HFox1 was used for preparing inoculum. A mixture of wheat and barley seeds (3:1, *v*/*v*) was soaked in water overnight. The following day, excess water was decanted and seeds were autoclaved and inoculated with *F. oxysporum* isolate. Inoculated grains were incubated for two weeks at 25 ± 2 °C and protected from sunlight. After 2 weeks, 6 g of the inoculum was placed 3–4 cm below the surface of sterile soil in each 12 × 10 cm pot. Control inoculum was prepared by applying sterile distilled water to the seed. Twenty-seven hemp seeds of cultivar Fibranova [49] were sown in pots with universal natural substrate Florafin (three seeds/pot in 3 replications). The pots were transferred to a room with controlled lighting and temperature at 25 ± 2 °C for 24 h. The first symptoms (wilting) of the disease were observed 3–4 weeks after sowing on more than half of the total 27 hemp seedlings. Symptoms were similar to those observed on plants in the field. Several plants were randomly selected to perform re-isolation and satisfy Koch’s postulates. Several pieces of stem cortex (1 to 2 cm pieces) samples with sterile blades were treated with 1% sodium hypochlorite solution for 5 min and rinsed three times with sterile water. Pieces were left to dry under hood flow and placed on half-strength potato dextrose agar (PDA) amended with streptomycin stock solution (Sigma-Aldrich, St. Louis, MO, USA) to inhibit bacterial growth.

## 3. Results

### 3.1. Fungal Isolate and Identification

Plants and roots with symptoms of external and internal browning (Figure 1) at the base of stems, interveinal chlorosis of leaves and death of shoots were collected in the plant ripening phase. Although according to the literature pulled-up roots infected with *F. oxysporum* show no external symptoms [33], we found a reddish-brown discoloration on the roots after removing the external bark layer (Figure 1b). 

Colony characteristics were observed after 7 days of incubation. Single-spore isolates produced septate cottony white to light pink aerial mycelium and colorless to light pink undersurface on potato dextrose agar (PDA) (Figure 2a). Smooth and hyaline hyphae were branched and septate. Macroconidia were fusiform to sickle-shaped with foot-shaped basal cells, elongated apical cells and three to five septa. In all our isolates three-septate macroconidia were commonly observed with an average size of 35.21 ± 7.8 × 3.52 ± 1.1 µm. Oval single-celled microconidia (Figure 2b) were produced only in false heads on short monophialides. The average size of microconidia was 8.91 × 2.17 µm.

### 3.2. Molecular Identification

Molecular identification of the pathogen was performed using two genes, the internal transcribed spaces (ITS) and the partial elongation factor 1-α. The internal transcribed spacer (ITS), which is widely used as a barcode in fungal community studies [50,51], does not provide species-level resolution for *Fusarium* species. Therefore, the TEF1 gene was included in the analysis, which is known to be a useful phylogenetic marker for *Fusarium*, frequently providing species-level discrimination [52]. An advantage of using the TEF1 gene for community studies is that it appears to be present as a single copy in *Fusarium*, allowing for more quantitative comparisons between species [52]. 

The internal transcribed spacers (ITS) sequences of the isolate HFox1 (524 bp) (GenBank accession No. OM475708) was amplified using primers ITS4/ITS5. The isolate was characterized by ITS sequencing and identified with 100% identity as *F. oxysporum*. Phylogenetic analyses (Figure 3) grouped HFox1 isolate with a high degree of sequence identity (99–100%) within the *F. oxysporum* complex. Isolate OM475708 showed 100% similarity with an *e*-value of 0 to the reference sequences of *F. oxysporum* f. sp. *melonis* (ISPaVe1018 GenBank accession number FR852562). On account of its polyphyletic nature, *F. oxysporum* forms a cluster with different groups of the *F. oxysporum* species complex [53]. 

The sequences of the same isolate HFox1 (634 bp) (GenBank accession number OR149071) were amplified using primers EF1/EF2 [49]. The isolate was identified with 100% identity as *F. oxysporum* f. sp. *melonis* (ISPaVe1018 GenBank accession number HE585984). Phylogenetic analyses (Figure 3) again grouped HFox1 isolate with a high degree of sequence identity (99–100%) within the *F. oxysporum* complex.

### 3.3. Artificial Inoculation

Three–four weeks after artificial inoculation, the inoculated seedlings showed stunted growth and wilt symptoms similar to those observed in the commercial fields. Stems of inoculated young plants showed the symptoms of external and internal browning at the base of stems and interveinal chlorosis of leaves, similar to the diseased plants collected in the field (Figure 4a,b). 

Three-quarters of the inoculated seedlings developed disease symptoms. Pathogen was successfully re-isolated from parts of all infected seedlings on PDA with the same morphological characters as before, which fulfilled Koch’s postulates. The control plants remained symptomless and healthy (Figure 4c). The isolates were stored in sterile distilled water at −18 °C and archived at the Department of Phytomedicine, Faculty of Agrobiotechnical Sciences Osijek, Josip Juraj Strossmayer University of Osijek for further use.

## 4. Discussion

In this study, according to cultural and conidial morphology, ITS4/ITS5 and EF1/EF2 sequence-based phylogenetic analysis and pathogenicity test, the pathogenic fungus was identified as *F. oxysporum* and confirmed to be the causal agent of Fusarium wilt in hemp (*C. sativa*). To the best of our knowledge, this is the first report of hemp Fusarium wilt causing *F. oxysporum* in Croatia. Considering that *F. oxysporum* has been reported not only in field crops [16,17], but also in many weed species [54], the presence of this pathogen could cause economically significant hemp production decreases. 

*Fusarium* is a typical soilborne genus, widely distributed and generally abundant in all types of soils around the world [55]. Some species complexes, such as the *F. oxysporum* species complex (FOSC), are considered ubiquitous, while the distribution of some other species depends more on climate conditions [56]. However, several other environmental factors, including soil characteristics, crops, cultural practices and human activities, may affect the diversity of *Fusarium* communities in soil, although the relative importance of these different parameters is poorly understood. Several soilborne fusaria, e.g., the different *formae speciales* of *F. oxysporum*, which are responsible for severe vascular wilts or root rot diseases in a wide range of crops of economic importance, are important plant pathogens [43]. It is important to identify the fusaria present at the species level, since they have different life cycles, host ranges, mycotoxin profiles, and climate preferences, and since they respond differently to control methods [57,58]. 

Species from the genus *Fusarim* are widely distributed and can be found in aquatic habitats, including seawater, river water [59], drinking water sources [60] and some populations seem to be particularly adapted to complex water distribution systems [61]. Several *Fusarium* species have also been reported as pathogens of marine animals [62,63]. It is significant that many *Fusarium* species are of clinical importance, causing, e.g., serious corneal infections [64] and invasive infections in immunocompromised patients [65,66]. *F. oxysporum* in Croatia is regularly detected on a large number of field crops (above all on corn and wheat, which are still the primary field crops in Croatia). In the study of Ivic et al. [16] the authors list the crops in Croatia on which they found the presence of *Fusarium* species, but do not determine them to form species (*formae specialis*) depending on what hosts they are to infect. A few years ago, the presence of this pathogen was also confirmed in soybeans [23]. A narrow crop rotation, along with favorable climatic conditions, certainly favors the development of the pathogen and its spread.

*F. oxysporum* attacks hemp seeds causing a decrease in quality. Fibranova [49] hemp cultivar was used in the pathogenicity test because our collected isolates were found in the field sown with this cultivar. The authors did not conduct a statistical analysis during the trial because the aim of this study was nothing but to simply determine the existence of a new pathogen which causes disease on hemp plants in a commercial field in the Republic of Croatia. Although is to be assumed that there may be differences in the susceptibility of cannabis varieties to this pathogen, the susceptibility of this variety was not the subject of this study. 

One of the ways to control the spread of this pathogen is certainly to create more resistant/tolerant or at least less susceptible varieties of important agricultural crops through breeding programs at the Osijek Agricultural Institute (corn, wheat, barley, soybeans). For this purpose, testing of materials (lines) in breeding programs was started through artificial infections in the laboratory, growth chamber and field experiments. Work is underway on testing soybean breeding material in which detected and determined isolates from hemp are used for cross-infection.

Using the described methodology, morphological characteristics and based on molecular analyses, we determined the new isolate HFox1. We confirmed the pathogenicity of the HFox 1 isolate to hemp by applying Koch’s postulates. The sequences [27,67] of isolate (GenBank accession no. OM475708 and OR149071) were compared with those available in the NCBI (National Center for Biotechnology Information, http://www.ncbi.nlm.gov/; accessed on 22 March 2022 and 12 July 2023) genetic database by using the Basic Local Alignment Tool (BLAST) algorithm showed 100% similarity with *e*-value of 0 to the reference sequences of *Fusarium oxysporum* (GU724513 and KX196809), as well as *F. oxysporum* (MK461973).

Hemp was regularly grown in eastern Croatia after, and especially between the two World Wars. By the middle of the twentieth century, the largest producers will remain concentrated in parts of Russia, Ukraine, Hungary and former Yugoslavia [68]. During the former Yugoslavia, hemp was one of the leading cultures, and the country itself was 3rd in the World in hemp production [69]. The area of Slavonia was significantly oriented towards the cultivation, processing and export of industrial hemp [70]. In the 1960s last century, hemp production was almost completely abandoned by passing laws that prescribed strict conditions for the industrial cultivation of hemp [70]. With cessation of production as was the accompanying processing industry. The industry hemp in Croatia has had a sharp increase in the past 5 years [69]. Production has been in constant increase, from 1560 ha in 2015 to 2476 ha in 2019 [71]. Upon the resumption of cannabis production in recent years in Croatia, the diseases have not been identified, or they have not been a problem nor have they been reported by producers. With the liberalization of the legislation, hemp production is expected to grow in the coming years. Narrowing the crop rotation could potentially increase the problem with diseases in general as well as the disease caused by *Fusarium*, especially in humid and warm springs and more susceptible varieties [21,33]. 

## 5. Conclusions

Although in Croatia hemp currently does not cover larger areas compared to other agricultural crops, in the last few years more and more farmers have decided to grow it. Legislation in the Republic of Croatia has been significantly improved in the last few years, but there is still resistance in practice due to the association of hemp with a plant that is used exclusively for making illegal drugs. The lack of processing capacities for different uses of hemp is certainly one of the factors for its slower spread. By educating producers, but also the entire public, there will certainly be an increase in the area under this valuable plant. This will also lead to a narrowing of the crop rotation, and consequently to an increase in problems with diseases such as Fusarium wilt, which attack a greater number of cultivated arable and vegetable crops. The changes in climatic conditions that we are witnessing in recent seasons will further emphasize the problem. This manuscript is only the first step in opening up the problem of hemp disease and the occurrence of Fusarium wilt, as well as a warning and guidelines for further work.

## Figures and Tables

**Figure 1 plants-12-03305-f001:**
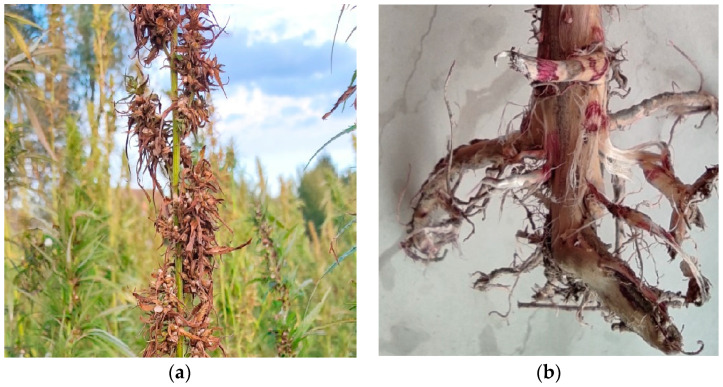
Fusarium wilt of hemp caused by *Fusarium oxysporum*. (**a**) Leaf symptoms of *F. oxysporum* on hemp in the field; (**b**) Root symptoms.

**Figure 2 plants-12-03305-f002:**
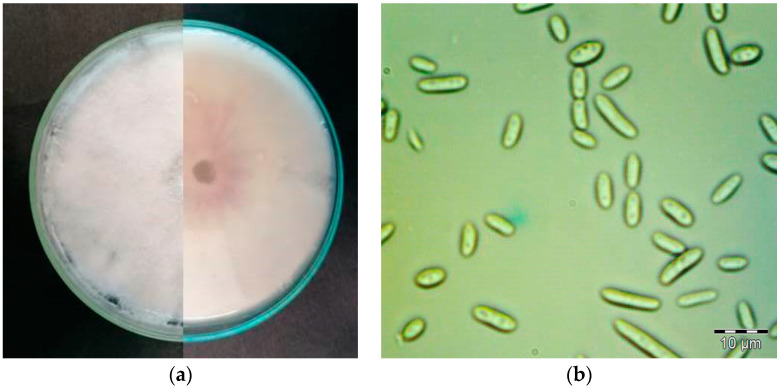
(**a**) 12-day-old colony of *F. oxysporum* on Potato Dextrose Agar (PDA) plate (90 mm), front and back; (**b**) microconidia of *F. oxysporum*.

**Figure 3 plants-12-03305-f003:**
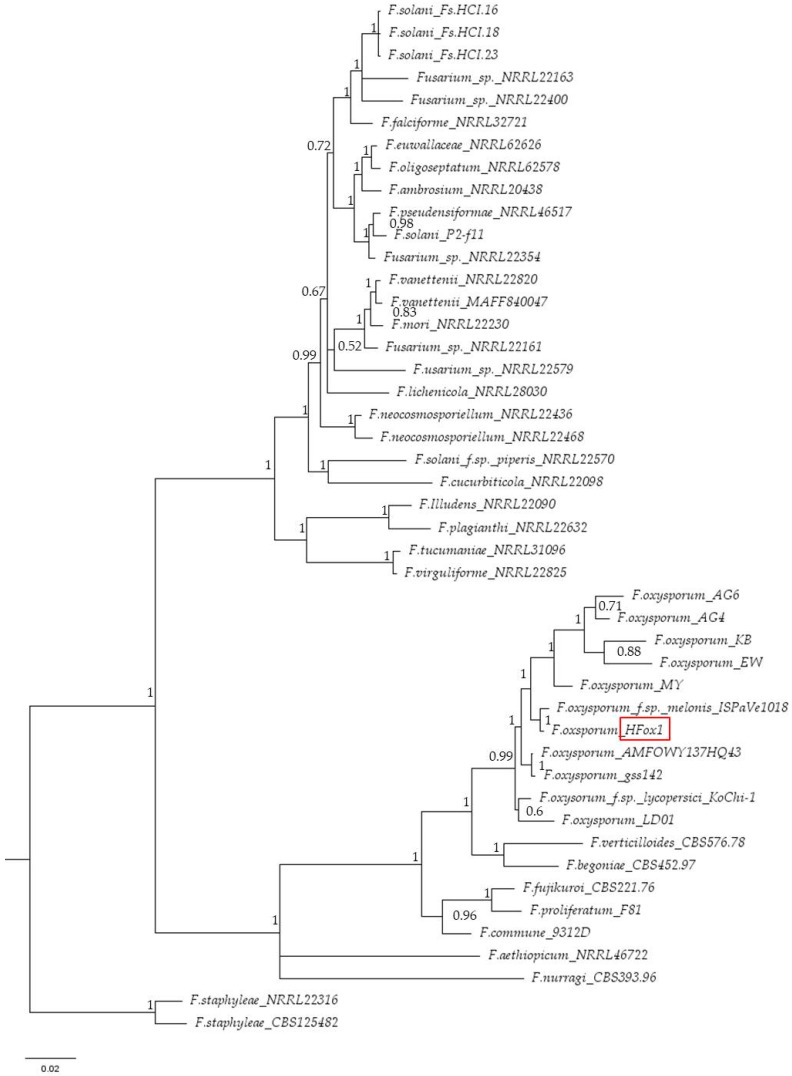
Phylogenetic tree inferred from Bayesian analysis on sequences of the concatenated dataset of the internal transcribed spacer (ITS) region and translation elongation factor (TEF1-α) of 46 representative species of *Fusarium*. Bayesian posterior probabilities are indicated beside nodes. *Fusarium staphyleae* (NRRL22316 and CBS 125482) obtained from GenBank were treated as the outgroup. Our isolate HFox1 (red box) is GenBank accession number OM475708 (ITS) and OR149071 (TEF1-α).

**Figure 4 plants-12-03305-f004:**
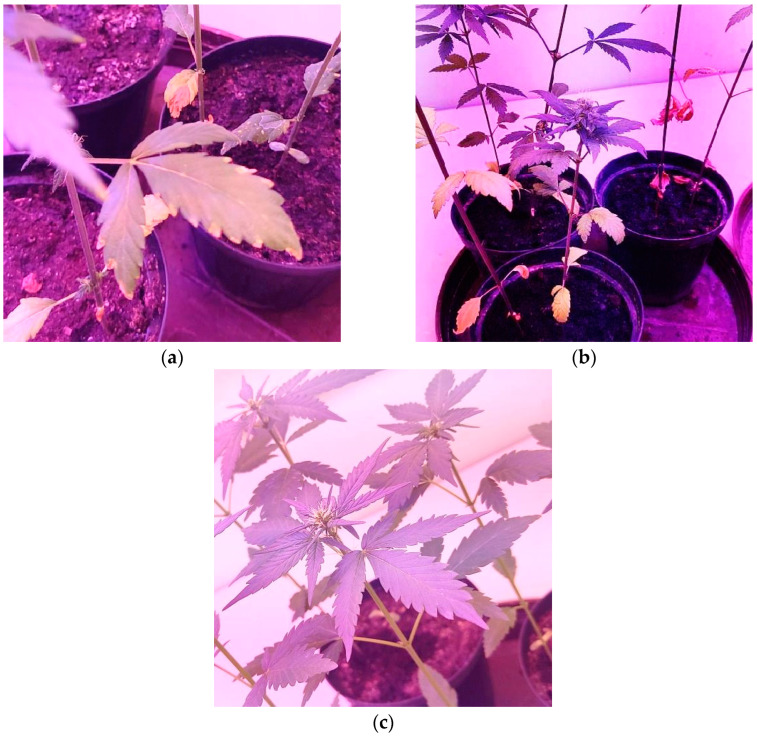
Symptoms of *F. oxysporum* artificial infection in laboratory conditions (**a**) Wilt symptoms begin with an upward curing of leaf tips; (**b**) Wilted leaves dry to a yellow-tan color and hang on the plant without falling off; (**c**) Control plants.

**Table 1 plants-12-03305-t001:** *Fusarium* species isolates and GenBank accession numbers used for the phylogenetic analysis.

Pathogen	Isolate	GenBank Accession No.	
		ITS	TEF-1α
*F. staphyleae*	NRRL 22316	AF178423	AF178361
*F. illudens*	NRRL 22090	AF178393	AF178326
*Fusarium* sp.	NRRL 22163	AF178394	AF178328
*F. ambrosium*	NRRL 20438	AF178397	AF178332
*F. solani Fusarium* sp.	NRRL 22354	AF178402	AF178338
*F. solani Fusarium* sp.	NRRL 22579	AF178415	AF178352
*F. plagianthi*	NRRL 22632	AF178417	AF178354
*F. solani* f. sp. *Piperis*	NRRL 22570	AF178422	AF178360
*F. commune*	9312D	DQ016191	DQ016253
*F. cucurbiticola*	NRRL 22098	DQ094301	AF178327
*F. batatas Fusatium* sp.	NRRL 22400	DQ094303	AF178343
*F. mori*	NRRL 22230	DQ094305	AF178358
*F. vanettenii*	NRRL 22820	DQ094310	AF178355
*Fusarium* sp.	NRRL 22161	DQ094311	AF178330
*F. neocosmosporiellum*	NRRL 22436	DQ094317	AF178348
*F. neocosmosporiellum*	NRRL 22468	DQ094318	AF178349
*F. lichenicola*	NRRL 28030	DQ094355	DQ246877
*F. falciforme*	NRRL 32721	DQ094503	DQ247041
*F. tucumaniae*	NRRL 31096	EF408523	GU170636
*F. virguliformae*	NRRL 22825	EF408542	GU170635
*F. aethiopicum*	NRRL 46722	FJ240308	GU170635
*F. vanettenii*	MAFF 840047	AB513852	AB513842
*F. oxysporum* f. sp. *lycopersici*	KoChi-1	AB675383	LC648711
*F. oxysporum* f.sp. *melonis*	ISPaVe1018	FR852562	HE585984
*F. euwallaceae*	NRRL 62626	KC691560	KC691532
*F. oligoseptatum*	NRRL 62578	KC691565	KC691537
*F. pseudensiformae*	NRRL 46517	KC691584	KC691555
*F. oxysporum*	AMFOWY137HQ43	KR047072	KR108763
*F. verticillioides*	CBS 576.78	KR071630	AB674287
*F. solani*	P2-f11	LC198903	LC198905
*F. oxysporum*	LD01	MH752745	MN026924
*F. nurragi*	CBS 393.96	MH862577	MW928840
*F. begoniae*	CBS 452.97	MH862660	MN533994
*F. staphyleae*	CBS 125482	MH863592	MW834282
*F. oxysporum*	MY	MG564287	MG674219
*F. solani*	Fs_HCI_16	MK393907	MK814525
*F. solani*	Fs_HCI_18	MK393909	MK814527
*F. solani*	Fs_HCI_23	MK393910	MK814528
*F. oxysporum*	AG6	MG564289	MG677117
*F. oxysporum*	KB	MG564290	MG677118
*F. oxysporum*	AG4	MG564293	MG677121
*F. oxysporum*	EW	MG564296	MG696756
*F. fujikuroi*	CBS 221.76	MW827608	MN534010
*F. proliferatum*	F81	MW995674	KU974246
*F. oxysporum*	gss142	MH290453	MH341211
*F. oxysporum*	HFox1	OM475708	OR149071

## Data Availability

All sequence data are available in NCBI GenBank following the accession numbers in the manuscript.

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
