# Peer review of "First Report of Hemp Fusarium Wilt Caused by Fusarium oxysporum in Croatia"

_plants, 2023, doi:10.3390/plants12183305_

Round 1
Reviewer 1 Report (Previous Reviewer 2)
This is a well written and informative manuscript. On line 35 you use the word bast. I am not sure what you mean, maybe best?
On your figures of the molecular identification of F. oxysporum, you should put the ID number for your isolate in the figure legend.
This is written clearly.
Author Response
We would like to thank you for your hard work and useful suggestions when reviewing our work.
On line 35 we use the word bast. Bast fiber (also called phloem fiber or skin fiber) is plant fiber collected from the phloem (the "inner bark", sometimes called "skin") or bast surrounding the stem of certain dicotyledonous plants. It supports the conductive cells of the phloem and provides strength to the stem. Some of the economically important bast fibers are obtained from herbs cultivated in agriculture, as for instance flax, hemp, or ramie, but bast fibers from wild plants, as stinging nettle, and trees such as lime or linden, willow, oak, wisteria, and mulberry have also been used in the past.
On both figures of the molecular identification of F. oxysporum, we put the ID number (GenBank accession numbers) for our isolate in the figure (3&4) legend. We have also marked our isolate on each of the figures.
Reviewer 2 Report (New Reviewer)
The manuscript titled "First report of hemp Fusarium wilt caused by Fusarium oxysporum in Croatia" presents an important finding regarding the occurrence of Fusarium wilt in hemp plants in Croatia. The study's findings could have significant implications for hemp production, especially given the potential economic impact of the pathogen. However, the manuscript requires a major revision to enhance its clarity, provide additional context, and address certain aspects that currently hinder its comprehension and scientific rigor.
Introduction and Contextualization: The introduction needs to be expanded to better contextualize the importance of hemp production in Croatia and globally. Additionally, it would be beneficial to introduce different applications of cannabis as well as hemp and drug-type cannabis. So, I suggest the following sentences:
"Cannabis has been widely cultivated due to its industrial (DOI: 10.3906/bot-1907-15), ornamental (10.3390/plants11182383), nutritional (10.3390/plants11233330), medicinal, and recreational (10.1016/j.biotechadv.2022.108074) potentials. From regulatory and application perspectives, cannabis plants are categorized based on the level of Δ9-tetrahydrocannabinol (THC), one of the most important phytocannabinoids (10.1146/annurev-arplant-081519-040203). Plants are generally classified and regulated as industrial hemp if it contains less than 0.3 % THC in the dried flower (this level varies by country) or drug-type with more than this threshold (10.1016/j.indcrop.2020.113026)."
Methods Description: The methods section lacks details necessary for reproducibility and scientific rigor. It is crucial to provide comprehensive information about the sampling protocol, the criteria used for selecting affected plants, and the specific symptoms observed. Further elaboration on how the isolates were obtained and maintained, as well as the conditions under which the artificial infection experiments were conducted, is necessary.
Pathogen Characterization: The section describing the pathogen's characterization needs to be more comprehensive. Provide more details about the methodology and rationale behind using the internal transcribed spacer and the partial elongation factor 1-α gene for identification. Additionally, describe the significance of these genes in identifying Fusarium species.
Results Interpretation: While the results indicate the presence of Fusarium oxysporum and the successful fulfillment of Koch's postulates, more discussion is required to explain the implications of these findings. Discuss the potential impact of this pathogen on hemp production, particularly in the local context of Croatia's climate and hemp varieties.
Discussion Enrichment: Expand the discussion section to include a broader context of Fusarium wilt in other crops and regions. Compare and contrast the findings with previous reports of Fusarium wilt caused by F. oxysporum. Additionally, address the implications of the pathogen's presence on a larger scale, including potential strategies for management and prevention.
Conclusion and Future Implications: The conclusion should succinctly summarize the significance of the study's findings, especially with regards to hemp production in Croatia. Highlight the potential consequences of the disease's spread and its possible impact on local agriculture. Suggest future research directions to better understand the epidemiology of this disease and strategies for mitigation.
Author Response
The authors thank you for your help in pointing out errors and areas for improvement in the manuscripts.
In the Introduction chapter, we have added papers that were suggested by reviewers to better show the importance of hemp production. We have improved the section that describes the methods and how the isolates were handled, as well as the way to perform the artificial infection. We explained the characteristics of the pathogen and described the importance of the genes used for molecular identification of the pathogen. We also clarified the confusion in the description of the artificial infection (the disputed sentence simply ended up completely wrong in this paper).
The authors made an effort to enrich the work in such a way that they followed the suggestions and pointed out shortcomings, which refer to the chapters Results and Discussion. A chapter was added to the Conclusion, in which we tried to briefly summarize the significance of the study's findings.
Minor errors pointed out have also been fixed (Molecular, World).
Reviewer 3 Report (New Reviewer)
The manuscript describes new symptoms on hemp in Croatia and presents enough evidence to conclude that the fungus Fusarium oxysporum is the causal agent for it. Because hemp is an important economic crop, and the newly described disease could have major impact on the cultivation in Croatia and other countries, I recommend the publication of the manuscript.
However, the manuscript could be improved with some rather minor modifications. Not all references listed in the Introduction are absolutely needed for the purpose of this article.
The English language should be checked.
The Introduction contains a lot of interesting information, but could be shortened by approximately one third for the purpose of this article.
Material and Method:
4.1 Fungal Strain Collection:
Some more information could be added for the location of the field with the symptomatic plants. What is the elevation, climatic conditions (min/max annual temperatures, annual precipitation)? How old were the symptomatic plants?
4.4 Pathogenicity Test:
This crucial part has to be explained in more detail. The inoculum (F. oxsyporum) was grown on wheat and barley seeds. But the next step is not clear: did the authors use macroconidia as the inoculum or did the use the ‘macroconidia-free’ barley/wheat seed mix (what I believe they did; but then why do they list the conidial concentration?). The authors added 6 g of the inoculum to the sterile soil, which only makes sense when it refers to the barley/wheat seed mix.
Then, hemp seeds were added to the inoculated soil. Based on the Results part, symptoms were detected two weeks after seeding, but this timeframe is crucial for the experiment and should be mentioned also in the Material and Methods part.
The authors state in the Discussion that no statistical analysis was done, because the aim of the study was to confirm that F. oxysporum is the causal agent of the observed disease symptoms, and not to study susceptibility of the host or virulence of the pathogen. Still, it would be interesting to know (and easy to observe), how many of the seedlings developed diseases symptoms, and if they all developed at the same time and on the same plant tissue. Also, the authors did not mention if the re-isolated F. oxysporum from all symptomatic plant tissues, or just from some selected ones.
Some minor errors (typos etc.).
E.g.line 126 Molecular (not Molecularal) Identification
Line 201: World (not Great) wars
Author Response
We would like to thank you for your hard work and useful suggestions when reviewing our work.
4.1 Fungal Strain Collection: We have accepted your suggestions and added more information for the location of the field with the symptomatic plants (elevation, climatic conditions (min/max annual temperatures, annual precipitation) as well as the stage for the symptomatic plants.
Subchapter 4.4 Pathogenicity test is part of chapter 4. Material and Methods. We mentioned the time when we noticed the symptoms also in the Material and Methods part. The paper stated that "After 2 weeks, 6 g of the inoculum was placed 3-4 cm below the surface of..." the first symptoms of wilting were observed after 3 weeks sawing.
We have added information on the number of seedlings that developed symptoms after inoculation, which were identical to those observed in the field during the first detection of the disease. We also added information on the type and number of plants with symptomatic plant tissue.
We also fixed some minor errors (Molecular, World).
Round 2
Reviewer 2 Report (New Reviewer)
All the comments have been addressed. The current version of the manuscript can be published in the journal.
Author Response
Thank you very much
This manuscript is a resubmission of an earlier submission. The following is a list of the peer review reports and author responses from that submission.
Round 1
Reviewer 1 Report
The manuscript by Duvnjak et al. is extremely well presented and organized and provides interesting introductory information that enriches the overall work presented. In the pdf file attached I provide minor comments on the style of writing and English spelling to improve clarity.

Author Response
Corrections:
Line 21 – revised English
Lines 36,37 – revise English suggested, but this is citation. It is not usual to make revision of cited sentence.
Lines 55,56 – revise English suggested, but this is citation. It is not usual to make revision of cited sentence.
Line 60 – authors ment time when infection starts and plant growth stage, we made correction.
Line 73 – made correction as suggested
Line 89 – „pathosystems“ – expression is part of cited sentence so we think that is not suitable to change it.
Lines 91,92 – revise English suggested, but this is citation. It is not usual to make revision of cited sentence.
Line 95 – „most economically important species“. Aslo part of citation, and in the first part of sentence is clear that is most economically importand amonf Fusarium species.
Line 98 – „discover“ – expression is part of cited sentence so we think that is not suitable to change it.
Line 108 – made correction as suggested
Line 117 – made correction as suggested
Line 119 – made correction as suggested
Line 145 – made correction as suggested
Line 148 – made correction as suggested
Line 173 – made correction as suggested
Line 176 – made correction as suggested
Line 180 – made correction as suggested
Lines 181-186 has been after one of the reviewers suggestion to explain one of his request
Line 209 – chane „after“ with „upon“
Line 192 – made correction as suggested
Line 212 – change expression with „legislation“
Lines 213-215 – revise English suggested, but this is citation. It is not usual to make revision of cited sentence.
Line 221 – made correction as suggested
Line 222 – made correction as suggested
Line 239 – made correction as suggested
Lines 161-169 - highlighted, but the remarks are not legible
Reviewer 2 Report
I have only one concern about the work presented. A recent publication of Fusarium experts, O'Donnell et al., 2022. DNA sequence-based identification of Fusarium: A work in progress. Plant Disease 106:1597-1609, indicated that the translation elongation factor 1-alpha (TEF-1) and 2 subunits (RPB1 and RPB2) of a DNA-directed polymerase were necessary to resolve Fusarium isolates at or near the species level. By contrast, the nuclear ribosomal internal transcribed spacer unit (ITS rDNA) are typically too conserved to distinguish Fusarium species. Unfortunately, in this manuscript, ITS4 and ITS5 were used to determine the species. Since the work in this paper was done before this new publication came out, the authors would not have the knowledge presented in the recent O'Donnell et al., publication. Prior to that publication, I would have accepted the molecular characterization of the fungus. All the other work in this document was very well done.
Other corrections:
line 37: available bast natural fiber. I am not sure if you mean best instead of bast.
Line 52: Fusarium should be in italics
Line 87: soil temperatures. [18] The citation should be before the period.
In the references, you need to only capitalize the first word of the title for a journal article. This needs to be corrected on citations 11, 13, 17, 18, 39, and 47.
Author Response
Corrections
line 37: available bast natural fiber. I am not sure if you mean best instead of bast.
Line 52: Fusarium should be in italics - correction made
Line 87: soil temperatures. [18] The citation should be before the period. - correction made
In the references, you need to only capitalize the first word of the title for a journal article. This needs to be corrected on citations 11, 13, 17, 18, 39, and 47. - correction made
Reviewer 3 Report
The present manuscript is a first report, and fulfills the critheria of a first report. However, it is delivered as a full article, and the content is not after my opinion. The introduction is too long and includes not relevant information. I think this manuscript should be shorted down to a first report format and submitted to a journal printing first reports.
Author Response
Authors present the manuscript as Communication, trying to fulfills the criteria of a first report. It is not delivered as a full article (Communication allows 10 pages). Authors first attempt to publish this manuscript was declined because one of the reviewer wrote that is too short (we need to make corrections in order to extend introduction, make a figure with phylogenetic tree, better explain molecular method, etc. This is the one and only reason why authors made the manuscript longer.
Round 2
Reviewer 3 Report
It seems that this article had been a first round before I got it. My advice was to change it to a First report and send it to a journal that print first reports. I think the content is suitable for that. I would not recommend it as a communication article. My advice from last time is still valid: the introduction is too long and with literature not needed.